# Potential Benefits of Continuous Glucose Monitoring for Predicting Vascular Outcomes in Type 2 Diabetes: A Rapid Review of Primary Research

**DOI:** 10.3390/healthcare12151542

**Published:** 2024-08-04

**Authors:** Radhika Kiritsinh Jadav, Kwang Choon Yee, Murray Turner, Reza Mortazavi

**Affiliations:** Faculty of Health, University of Canberra, Canberra, ACT 2617, Australia; radhikashah2529@gmail.com (R.K.J.); kwang_choon.yee@canberra.edu.au (K.C.Y.); murray.turner@canberra.edu.au (M.T.)

**Keywords:** diabetes mellitus, T2DM, glycaemic variability, vascular complications, glucose fluctuations, glucose excursions, continuous glucose monitoring, CGM, rtCGM, isCGM

## Abstract

(1) Background: Chronic hyperglycaemia is a cause of vascular damage and other adverse clinical outcomes in type 2 diabetes mellitus (T2DM). Emerging evidence suggests a significant and independent role for glycaemic variability (GV) in contributing to those outcomes. Continuous glucose monitoring (CGM) provides valuable insights into GV. Unlike in type 1 diabetes mellitus, the use of CGM-derived GV indices has not been widely adopted in the management of T2DM due to the limited evidence of their effectiveness in predicting clinical outcomes. This study aimed to explore the associations between GV metrics and short- or long-term vascular and clinical complications in T2DM. (2) Methods: A rapid literature review was conducted using the Cochrane Library, MEDLINE, and Scopus databases to seek high-level evidence. Lower-quality studies such as cross-sectional studies were excluded, but their content was reviewed. (3) Results: Six studies (five prospective cohort studies and one clinical trial) reported associations between GV indices (coefficient of variation (CV), standard deviation (SD), Mean Amplitude of Glycaemic Excursions (MAGE), Time in Range (TIR), Time Above Range (TAR), and Time Below Range (TBR)), and clinical complications. However, since most evidence came from moderate to low-quality studies, the results should be interpreted with caution. (4) Conclusions: Limited but significant evidence suggests that GV indices may predict clinical compilations in T2DM both in the short term and long term. There is a need for longitudinal studies in larger and more diverse populations, longer follow-ups, and the use of numerous CGM-derived GV indices while collecting information about all microvascular and macrovascular complications.

## 1. Introduction

Type 2 diabetes mellites (T2DM) is a prevalent chronic disease globally associated with a wide range of morbidities including macrovascular (ischaemic heart disease, stroke, and peripheral artery disease) and microvascular (nephropathy, neuropathy, and retinopathy) complications [1,2]. The percentage of glycated haemoglobin A (HbA1c) reflects the overall glycaemic exposure of haemoglobin over the previous 2–3 months [3,4]. In the late 1980s and 1990s, two landmark studies (Diabetes Control and Complications Trial (DCCT) and UK Prospective Diabetes Study (UKPDS)) demonstrated HbA1c’s role in assessing the effectiveness of long-term diabetes control and predicting future vascular complications [5,6]. Other studies demonstrated direct relationships between HbA1c levels and risk of vascular complications [7,8,9]. The use of HbA1c has since become the gold standard for assessing long-term disease management in both primary care and clinical studies [4,10]. However, other studies reported that lowering HbA1c levels intensively (i.e., approaching those of non-diabetic individuals) compared with standard practice did not reduce the risks of cardiovascular disease or all-cause mortality, and instead, increased the risk of severe hypoglycaemia [11,12].

Despite being a very useful tool in guiding diabetes management, HbA1c is unable to identify short-term glycaemic variability (GV) [13] or assess glycaemic situations in haemoglobinopathies [14], iron deficiency anaemia [14], and pregnancy [15]. These limitations may account for the wide variations in microvascular and macrovascular complications in patients who met the target HbA1c values for glycaemic control [13].

In 2014, researchers from the large international ADVANCE trial reported a direct association between visit-to-visit variability (VVV) in HbA1c levels and increased risks of macrovascular complications and death in patients with T2DM [16]. They also found a significant association between the VVV of fasting blood glucose and risks of vascular events (both macro- and microvascular events).

It appears that the mechanisms responsible for the development of vascular lesions in patients with diabetes are multifactorial and complex. Accordingly, the glycation of cellular proteins [3] and membrane lipids [17], proinflammatory responses [14], oxidative stress [18], and coagulopathies [19] have been suggested in addition to other risk factors for atherosclerosis such as hypertension, high LDL-cholesterol, obesity, age, and smoking. In vitro studies demonstrated that tissue injuries from hyperglycaemia are both time dependent and glucose concentration dependent, and significant cellular injury could be detected within as short as 24 h [20,21]. A case-control study of 21 patients with T2DM suggested the existence of an oxidative damage-triggering role for blood glucose fluctuations (especially postprandial), which could damage the blood vessels independent of sustained hyperglycaemia [22]. To make the matter more complex, treatment-related hypoglycaemia was also linked to adverse complications such as cardiac arrhythmia and death in different patient populations, especially among old patients, those with a high burden of comorbidities, or patients with impaired cognitive ability [19]. Given the fact that HbA1c does not have the capacity to measure short-term glucose excursions or detect hypoglycaemic events, there is a need for other tests which can provide this kind of supplementary but essential information.

GV may be a valuable independent factor in assessing the risk of vascular complications in patients with diabetes mellitus [1,23,24]. Over the past few decades, many GV indices (metrics) have been introduced with different applications. Some indices include glucose standard deviation (glucose SD), glucose coefficient of variation (glucose CV), Time in Range (TIR), Time Above Range (TAR), Time Below Range (TBR), Mean Amplitude of Glycaemic Excursions (MAGE), Mean of Daily Differences (MDD), Continuous Overlapping Net Glycaemic Action (CONGA), Mean Absolute Glucose (MAG), Glucose Management Indicator (GMI), and CV or SD of VVV HbA1c [1,25]. Table 1 summarises the definitions and clinical applicability of some of the commonly used indices. More information regarding these and other indices can be found in the references provided.

Before the 2000s, the self-monitoring of blood glucose (SMBG) using capillary blood, or pathology lab testing (using venous blood) were the only practical ways of monitoring glucose levels. Although these inconvenient methods are still commonly in use, the introduction of continuous glucose monitoring (CGM) systems to medical devices markets in 1999 and the early 2000s created new opportunities for better glucose management as well as more convenience for patients. Currently, the real-time CGM (rtCGM) and intermittently scanned CGM (isCGM) (also called flash glucose monitoring) are the two main types of commercially available CGM systems [32]. These systems allow for the measurement of glucose of the interstitial fluid following the insertion of a sensor into an arm or the abdominal skin with glucose measurements being undertaken automatically every 5–15 min [33,34]. Currently, these systems are more commonly used for the management of type 1 diabetes mellitus (T1DM); however, they have not been widely adopted for T2DM because of the lack of strong evidence for their clinical benefits and cost effectiveness [33,34].

The aim of this rapid review was to explore the existing evidence about the usefulness of CGM-derived GV indices in predicting vascular and other clinical outcomes in patients with T2DM.

Our research objectives were as follows:

Objective 1: To explore associations between blood glucose excursions (measured through GV indices) and short- or long-term vascular and clinical complications in patients with T2DM.

Objective 2: To investigate short- or long-term prognosis for patients with T2DM whose treatments were informed by CGM data compared with those patients whose treatments were devised solely based on HbA1c and glucose levels.

## 2. Materials and Methods

This rapid review of the literature was performed according to the *a priori* protocol described below. The reporting of this review was guided by the standards of the Preferred Reporting Items for Systematic Reviews and Meta-Analyses (PRISMA) 2020 Statement [35].

### 2.1. Data Source and Literature Search

The literature search was completed by two reviewers (RKJ and MT) including an expert systematic review librarian in December 2023. The Cochrane Library, MEDLINE, and Scopus were searched for relevant articles. A decision was made to limit the search to post-1999 as different CGM systems were introduced into diabetes clinical practice in 1999 and the early 2000s [32]. Only studies published in the English language were included.

For the database search, a range of search terms (including but not limited to “type 2 diabetes”, “continuous glucose monitoring”, “flash glucose monitoring”, “haemoglobin A1c”, HbA1c, macrovascular, microvascular, “heart disease” and stroke) were utilised in groups to cover the concepts of Population (patients with T2DM), Intervention (use of CGM), Comparison (use of HbA1c), and Outcomes (vascular complications). A full record of the search strategy and the number of results found from databases is included in Appendix A. In addition to the database search, the reference lists of included articles were checked to identify additional articles.

Following the database search, all articles were uploaded to the Covidence systematic review software for deduplication and the article selection process [36]. All reviewers contributed to title and abstract screening, and KCY, RKJ and RM carried out the full-text screening and information extraction from the included articles. Quality assessment of the included studies was undertaken by KCY and RM along with cross-checking of all the results. Conflicts were resolved by discussions or by RKJ.

### 2.2. Eligibility Criteria

#### 2.2.1. Type of Participants

Studies that recruited persons 18 years of age or older with T2DM, regardless of insulin administration, were included. Studies including only T1DM, gestational diabetes, children, or animals were excluded. In studies that recruited both T2DM patients and other types of diabetes, only data pertaining to patients with T2DM were extracted and analysed.

#### 2.2.2. Type of Intervention

Studies utilising CGM were included, and any studies which did not use CGM were excluded. As a comparison intervention, studies using HbA1c measurements with or without the occasional testing of blood glucose levels were included.

#### 2.2.3. Type of Studies

Randomised controlled trials, non-randomised trials, cohort studies, and case-control studies were considered eligible for inclusion. Other study designs such as cross-sectional studies, case series, case reports, case studies, and expert opinion papers were all excluded due to being generally considered of lower quality [37].

#### 2.2.4. Type of Outcomes

This review considered short-term and long-term macrovascular (ischaemic heart disease, stroke, and peripheral artery disease), microvascular (endothelial cell damage, nephropathy, neuropathy, and retinopathy) outcomes, and abnormalities in the heart’s electrical system. Studies were excluded if they only used HbA1c testing as a surrogate measure for clinical outcomes but did not directly measure clinical outcomes.

### 2.3. Risk of Bias

Quality of observational studies was assessed using the Newcastle–Ottowa Scale (NOS) [38]. The risk of bias for the randomised clinical trial was assessed using the 2019 version of the Cochrane Risk-of-Bias tool for randomised trials (RoB 2) [39].

### 2.4. Data Extraction

Data were extracted from the included studies by two reviewers and cross-checked by a third reviewer. A bespoke data extraction template was developed by the authors based on the aims of the review. Data were extracted based on general information, baseline participant characteristics, intervention, comparator, outcome of interest, main findings, and sponsorship. More specifically, the following data were extracted from the included studies: author, year of publication, geographic location, aims of the study, setting, study design, participants’ demographic data, duration of diabetes, comorbidities, macrovascular complications, microvascular complications, instrument (CGM), details of intervention, GV indices, duration of CGM, HbA1c comparator, follow-up period, findings, and sponsorship details.

## 3. Results

The preliminary searches identified 3544 records to which one additional article was added through reference list checking (total number = 3545). Of those, after deduplication, title and abstract screening and full-text screening, only six articles were included. The process of screening, exclusions (with reasons for them) and inclusions is reported in Figure 1 as per the PRISMA 2020 flow diagram. A full record of database searches has been provided in Appendix A.

### 3.1. Characteristics of the Included Studies

The main characteristics of the included articles are presented in Table 2 with more details presented in Appendix A. Five of the six included studies had a prospective cohort study design [40,41,42,43,44], while one was an open-label randomised controlled clinical trial [45]. The two articles by Lu et al. [40,43] are outputs of a single large study (the INDices of continuous Glucose monitoring and adverse Outcomes of diabetes (INDIGO) study), but since they have different aims and objectives, they were considered two separate studies in this review. Of the six included studies, three were undertaken in China [40,41,43], one in Japan [42], one in Denmark [44], and one in the UK [45]. In total, these studies recruited 7746 unique participants with individual study participant numbers ranging from 21 to 6225. The included studies were similar in terms of mean or median age of participants (ranged between 62 and 67), sex (majority male), and the type of diabetes (T2DM), but they were considerably different in terms of duration of diabetes (ranging from 2 to 18 years), median follow-up period (days to weeks to several years), or clinical outcomes (a range of microvascular and/or macrovascular complications). Settings were also varied from outpatient diabetes clinics to different types of hospitals. The aims and findings of the studies are summarised in Table 3.

### 3.2. Quality of the Included Studies

Based on the NOS, which uses a star-based scoring system, studies with seven or more stars were identified as high quality, whereas those with less than seven stars were assessed as moderate or low quality (without making a distinction between the two). Accordingly, of the five prospective cohort studies, only the study by Mita et al. is of high quality, whereas the other four studies are of medium or low quality. Table 4 presents the quality assessment results for each study.

Overall, the quality of the only randomised clinical trial was determined as high (see Appendix A). However, this study did not have sufficient power to effectively assess the vascular outcomes.

### 3.3. Associations between GV and Short-Term Vascular Complications of T2DM

Only one study (Su et al.) examined the short-term (days to weeks) associations of GV and vascular complications in patients with T2DM. This study recruited 759 patients who had been admitted to the hospital for elective percutaneous coronary intervention (PCI) following experiencing non-ST segment elevation acute coronary syndrome (NSTE-ACS) [41]. The subjects wore CGM sensors for 24–72 consecutive hours. Using multiple regression analysis, the researchers showed that patients with MAGE values ≥3.9 mmol/L compared to those with MAGE <3.9 mmol/L were at twice the risk of in-hospital major adverse cardiac events (MACEs) including all-cause mortality, new-onset myocardial infarction, acute heart failure, and stroke (OR 2.024; 95% CI 1.105–3.704) after adjusting for age, sex, CVD risk factors, and complications. The study also reported that using area under the receiver operating characteristic curve (AUROC), MAGE compared with HbA1c was a better predictor of in-hospital MACE (AUROC for MAGE 0.608, 95% CI 0.524–0.692, *p* = 0.012; AUROC for HbA1c 0.556, 95% CI 0.475–0.637, *p* = 0.193). The *p* value of the AUC for HbA1c indicates that in predicting short-term adverse vascular outcomes in this cohort of patients with T2DM, the test had a discriminative ability slightly better than flipping a coin. The study did not report the measurement of any other GV indices.

### 3.4. Associations between GV and Long-Term Vascular Complications of T2DM

The study by Ajjan et al. [45] was an open-label randomised controlled trial undertaken in nine secondary hospitals in the UK which sought to compare the benefits of using isCGM with self-monitoring of blood glucose (SMBG) in improving TIR, hypoglycaemic exposure (glucose <3.9 mmol/L), HbA1c, clinical outcomes (glycaemic emergencies, MACE and all-cause mortality), quality of life, and cost-effectiveness for a particular group of patients with T2DM. The study involved 141 patients with diabetes who recently had a myocardial infarction (MI) and were being treated with insulin and/or a sulphonylurea before the hospital admission. Upon recruitment to the study, participants were randomly assigned to either the isCGM group (n = 69) or the control group (SMBG) (n = 72). The primary outcome measure was TIR for the period of 76–90 days after randomisation. Secondary outcome measures included TIR during the period of 16–30 days following randomisation, time in hypoglycaemia per day during the two above-mentioned periods, and HbA1c and Quality of Life measures both at day 91 following randomisation. The exploratory outcome measures included hypoglycaemic emergencies, MACE, and death from all causes (the small sample size was the main reason for putting these outcomes into the exploratory category). The study found a significant reduction in hypoglycaemia and a marginal increase in TIR in the isCGM arm compared with the SMBG arm over three months. A clinically significant reduction in HbA1c at three months was observed in both arms of the study. There was no significant difference in vascular complications between the two groups. However, it should be noted that the study was not primarily designed to study vascular outcomes due to the small sample size and hence low statistical power.

The two related articles by Lu et al. (both published in 2021) were derived from the same study. This was a longitudinal cohort study that enrolled 6225 T2DM patients from local affiliated hospitals in China between 2005 and 2015. Participants’ demographics, clinical data and laboratory test results were recorded at hospitalisation, and then they were fitted with a blinded CGM system for three days to assess their baseline GV indices. The primary outcome of this study was mortality, in which CVD-related mortality was differentiated from other causes of death. The participants were followed up until either death occurred or 31 December 2018, whichever came first. The study had a median follow-up of 6.9 years during which 838 deaths were recorded. The mortality data were obtained through linking the personal identifiers of the patients with a central database (the Shanghai Municipal Centre for Disease Control and Prevention) [40,43]. In the first article [40], TIR was analysed in two ways. One was to categorise it into four groups including TIR > 85%, TIR = 71–85%, TIR = 51–70%, and TIR ≤ 50%, and the second way was to use TIR as a continuous variable. The multivariable-adjusted HRs for all-cause mortality for different TIR categories were as follows: 1.00 for TIR > 85% (reference group), 1.23 (95% CI 0.98–1.55) for TIR category of 71–85%, 1.30 (95% CI 1.04–1.63) for TIR of 50–70%, and 1.83 (95% CI 1.48–2.28) for TIR ≤50% (*p* for trend < 0.001). For CVD mortality, the HRs were 1.00 for TIR > 85% (reference group), 1.35 (95% CI 0.90–2.04) for TIR of 71–85%, 1.47 (95% CI 0.99–2.19) for TIR of 50–70%, and 1.85 (95% CI 1.25–2.72) for TIR ≤50% (*p* for trend = 0.015). The study also reported a statistically significant continuous relationship for each 10% decrease in TIR with the all-cause mortality (HR 1.08 (95% CI 1.05–1.12)) but not with the CVD mortality (HR 1.05 (95% CI 1.00–1.11)). In addition, the study also found that patients whose baseline HbA1c levels were <6% or ≥8% had higher CVD mortality compared with those with HbA1c of 6.0–6.9%. TIR was also negatively associated with HbA1c levels, the history of CVD, and use of hypertension medicines, aspirin, and statins.

In the second article by Lu et al. [43], the authors examined the relationships between HbA1c, glucose CV, TIR, TAR7.8, TAR10, TAR13.9, TBR3.9, and TBR3 in relation to all-cause mortality. This study included 6090 of the 6225 subjects from the INDIGO study (no details provided for the exclusions). A total of 815 deaths were recorded for the study period. Patients were categorised into four groups depending on their HbA1c levels (including groups with HbA1c of <6.0%, 6.0–6.9% (the reference group), 7.0–7.9%, and ≥8.0%). Independently from HbA1c grouping, the patients were classified into three tertiles according to their glucose CV (the low, the medium, and the high glucose CV tertiles). The researchers then used a Cox proportional hazard model to estimate the risk of all-cause death associated with different levels of HbA1c and other glycaemic metrics across the tertiles of glucose CV. Accordingly, it was found that among patients in the lowest and middle tertiles of glucose CV, an HbA1c level of 8.0% or higher was associated with a 136% (HR 2.36, 95% CI 1.46–3.81) and 92% (HR 1.92, 95% CI 1.22–3.03) increased risk of all-cause mortality, respectively, when compared to an HbA1c level of 6.0–6.9% (the reference group) in the fully adjusted model. However, in the highest tertile of glucose CV, there was no significant difference in the risk of all-cause mortality among the four HbA1c groups. This indicates that among the diabetic patients studied, HbA1c was not a good predictor of long-term mortality among patients in the highest tertile of baseline GV determined by glucose CV. However, contrary to HbA1c, each 10% decrease in TIR was associated with a 12% (HR 1.12, 95% CI 1.04–1.20) increase in the risk of all-cause mortality, while every 10% increase in the percentage of time spent above 7.8 mmol/L, above 10 mmol/L, and above 13.9 mmol/L was associated with an 11% (HR 1.11, 95% CI 1.04–1.19), 11% (HR 1.11, 95% CI 1.04–1.18), and 14% (HR 1.14, 95% CI 1.04–1.25) increased risk of all-cause mortality among the patients in the highest tertile of glucose CV, respectively. These findings suggest that in patients with T2DM and high-glucose fluctuations, TIR and TAR compared with HbA1c may be better predictors of long-term mortality.

The study by Mita et al. (2023) [42] is a sub-analysis of a larger cohort study undertaken in 34 outpatient settings across Japan which sought to assess the relationship between CGM-derived GV metrics and the occurrence of composite cardiovascular outcomes prospectively. The primary aim of the study was to examine the relationship between the baseline TIR and glucose CV, and changes in the intima–media thickness (IMT) and grey-scale median (GSM) in carotid arteries (both are ultrasonographic indicators of the progress of atherosclerosis) during 104 weeks of the study (two years). The secondary aim was to evaluate the association of other glucose variability indices at baseline and HbA1c with changes in IMT and GSM [42]. The participants in this study (n = 600) were selected from consecutive outpatients with T2DM (n = 1000) who met the inclusion criteria (aged ≥30 years and ≤80 years, did not have a symptomatic cardiovascular disease at the inclusion, glucose management regime was stable for the previous six months, and had baseline data for CGM, HbA1c, and carotid ultrasound images). CGM data were collected during the middle 8 days of the 14-day lifetime of the CGM sensor. There was no significant change in the mean HbA1c values for the baseline and the end of study (i.e., 104 weeks), but the glucose CV increased slightly but significantly during this period (25.8% ± 5.9% at the baseline, 26.6% ± 5.8% at 104 weeks; *p* < 0.001). Also, a significant increase in the mean IMT was observed (0.759 ± 0.153 mm at the baseline vs. 0.773 ± 0.152 at 104 weeks; *p* < 0.001) and thickened-lesion GSM (units) (43.5 ± 19.5 units at the baseline vs. 53.9 ± 23.5 units at 104 weeks; *p* < 0.001). The study found that TIR and glucose CV were associated with increased GSM and thickened-lesion GSM (independent from HbA1c). However, glucose CV and TIR were not associated with the increase in IMT or CCA-max-IMT. HbA1c was not associated with changes in the mean IMT, mean GSM or thickened-lesion GSM after adjusting for multiple testing.

Our last included study by Andersen et al. was a Danish prospective cohort study which explored the associations between glucose excursions and hypoglycaemia with cardiac arrhythmias in patients with T2DM treated with insulin [44]. Twenty-one patients with T2DM (mean age 67 years (SD = 10 years), mean duration of diabetes 18 years (SD = 8 years), mean HbA1c 6.8% (SD = 0.4%)) were recruited from diabetes outpatient clinics to this one-year study. All participants had one or more microvascular complication (nephropathy, retinopathy, or neuropathy), while four of them had also a macrovascular complication (two had coronary artery disease, one had cerebrovascular disease, and one had peripheral artery disease). Patients with history of cardiac arrhythmia, implanted cardiac defibrillator or pacemaker, severe heart failure, cardiac structural abnormalities or thyroid dysfunction were excluded from the study. During the study period, patients had an average total of 118 ± 6 days of CGM (blinded system) per individual spread over the 12 months. Three weeks prior to the start of CGM data collection, each patient had received an implantable cardiac monitor (ICM) subcutaneously, and a home monitoring system was established to facilitate the daily automated transmission of cardiac data (heart rate and arrhythmia) to the research team. The study found that time in hypoglycaemia was longer at night compared to daytime (median 0.7% and 0.4% respectively, *p* was not reported), while the severity of hypoglycaemia was slightly more during the daytime. The mean glucose CV during the daytime (27.9%, 95% CI 25.0–35.2%) was higher than during the nighttime (26.3%, 95% CI 23.0–33.7%) (no *p* value reported). Over the study period, 12 out of the 21 patients (57%) experienced clinically significant arrhythmias with the number of episodes ranging from 1 to 522 for individual patients. Asymptomatic cardiac arrhythmias occurred more frequently at night compared to the daytime (incident rate ratio (IRR) 4.22 [3.48–5.15]. Parallel CGM data were recorded for 29% of arrhythmias. No incidence of concomitant hypoglycaemia and arrhythmias was recorded during the daytime. The incidence rate of arrhythmias had positive associations with CV and SD during the nighttime but negative associations during the daytime. No significant difference was observed for incidence rates of arrhythmias during hyperglycaemia compared to euglycaemia either for the daytime or nighttime (IRR 1.14 [0.61–2.16] and IRR 0.89 [0.47–1.67], respectively). The study concluded that among patients with T2DM receiving insulin, cardiac arrhythmias were common and were associated with glucose excursions. No strong association was observed between hypoglycaemia and arrhythmias.

## 4. Discussion

The findings of this rapid review revealed that the direct relationships between glucose fluctuations (measured through GV indices) in patients with T2DM and short- or long-term vascular complications have not been extensively studied. However, the existing limited evidence is suggestive of both short-term and long-term associations. Regarding the short-term relationships, a positive association was reported for MAGE and in-hospital major adverse cardiac events [41]. The long-term prognostic relevance of GV has been studied relatively more with reports of associations between decreased baseline TIR and increased long-term risk of all-cause and CVD mortality [40], TIR and glucose CV with ultrasonographic atherosclerotic changes in the carotid artery wall [42], and glucose CV and glucose SD with the incidence rate of arrhythmias [44].

The study by Ajjan et al., although underpowered in some respects, was able to indicate that the use of is-CGM compared with SMBG for three months was associated with lower hypoglycaemic exposure and marginally increased TIR in the is-CGM group. These findings look promising and indicate the need for a direct study of clinical outcomes. We did not find a study that compared the use of CGM with HbA1c in improving vascular and other clinical outcomes. In fact, in some studies, it was conservatively decided that no changes were to be made to therapeutic regimens in response to the CGM results [40,43]. The study by Ajjan et al. was ranked as high quality in our rapid review; however, given the specifications of the included participants, it is recommended to take caution in extrapolating the results to wider groups of patients with T2DM.

Another comparison among GV indices and HbA1c was made in the study by Su et al. [41] where the researchers reported a better short-term predictive ability for MAGE compared with HbA1c for predicting in-hospital (short-term) cardiac adverse events for patients with T2DM and NSTE-ACS.

There is growing evidence suggesting that blood glucose fluctuations and hypoglycaemia have independent roles in causing or worsening vascular damage in patients with diabetes through multiple mechanisms such as oxidative damage and prothrombotic states [22,46]. The rapid fluctuation of blood glucose can be observed within hours or days after acute events (e.g., acute myocardial infarction), which could deteriorate the clinical prognosis for patients regardless of sustained hyperglycaemia. Studies have suggested that glucose fluctuation during these peri-ischaemic periods can cause vascular damage and contribute to the worsening of complications in short term [47,48]. Given the nature of HbA1c formation, it is unlikely that this test can be effective in predicting this acute risk, but a CGM system may be more beneficial in doing so.

There were common themes in the included studies which contributed to their moderate to low quality. First, most of the studies used very limited time to undertake CGM for calculating GV indices (ranged between 1 and 8 days among four studies) [40,41,42,43]. This is shorter than the 14 days of active CGM use recommended by the American Association of Clinical Endocrinology Clinical Practice Guidelines [49]. Second, most of the studies used a very specific population with conditions (such as MI), which affects the applicability of the results to the wider populations with T2DM. Third, in most of the studies, CGM was only undertaken at the baseline, and its results were compared with clinical outcomes years later. Only one study repeated CGM (three months after the initial measurements), but it did not provide a detailed analysis on the effects of any changes in GV indices measured between the two time points. The studies also did not follow the patients thoroughly during the study period in which they might have developed new clinical conditions which could have changed the clinical course of their diabetes dramatically. This issue should be addressed by longitudinal studies in the future. Forth, despite the availability of many GV indices, the included studies had limited their use to a handful of the indices (glucose CV, glucose SD, TIR, TAR, TBR, and MAGE). This might be due to the lower familiarity of the researchers, physicians, or patients with other GV metrics. Qualitative research (e.g., surveys) might be of value in obtaining insight into this matter. A deeper knowledge of clinicians, nurses, diabetes educators, and patients about the relationships of GV indices and diabetes complications can influence the future of diabetes management through more effective personalised medicine [33].

During our article screening process, several cross-sectional studies were excluded to comply with the search protocol (usually, these kinds of studies provide lower-level evidence compared with RCT, cohort studies or case-control studies). However, reading the excluded studies, some insights were achieved. Accordingly, several studies had assessed the association of GV and diabetic neuropathy and retinopathy [23,50,51,52,53,54]. The findings from those cross-sectional studies were mixed, but in general (and without being systematically assessed for their quality), they reported associations between different vascular complications and greater GV. Also, a large proportion of those studies was conducted in Asian countries (e.g., China, Korea, Japan, Taiwan), which usually have low diversity in terms of the participants’ race and ethnicity. Therefore, higher-level studies in more diverse populations are needed to be conducted to draw more applicable conclusions.

Based on our review of the literature (and reading through the articles that were not included), GV indices may be beneficial in predicting vascular complications in T2DM, and they might even be better than HbA1c in predicting short-term complications. However, due to the scarcity of evidence, we are unable to recommend that GV indices should be used in T2DM management in any situation. There is a need for high-quality prospective research and meta-analyses to find an answer for this question.

Our study had some strengths. Firstly, we focused on finding evidence for the association of GV with vascular outcomes (clinical). This was in contrary to most studies which had studied the associations of GV and HbA1c as a surrogate marker for clinical outcomes. Secondly, the processes including data extraction from the included studies and quality assessment were performed at least by two researchers independently with conflict resolutions made through consensus or by the third reviewer.

There are some limitations to our study as well. Firstly, to streamline the rapid review (as per our protocol), the title and abstract screening was completed only by one researcher (although all researchers were involved to different degrees). Secondly, we only included articles published in the English language, which may have caused us to miss significant articles published in non-English languages. Thirdly, because of the nature of the review being a rapid review, we only searched three databases. This means we might have missed articles indexed in other databases but not in those three.

## 5. Conclusions

This rapid review was able to find several lines of evidence to suggest the potential usefulness of isCGM- or CGM-derived GV indices in predicting short- or long-term vascular complications of T2DM. However, since most of the evidence came from studies with moderate to low quality, or undertaken in a very specific group of patients, their results must be interpreted with caution. The inclusion of CGM or isCGM into treatment protocols in T2DM might be useful in improving diabetes outcomes, but more studies are needed before the widespread adoption of the technology. From our point of view, well-designed longitudinal research needs to be conducted in larger groups of people from diverse backgrounds to find more credible and robust evidence.

## Figures and Tables

**Figure 1 healthcare-12-01542-f001:**
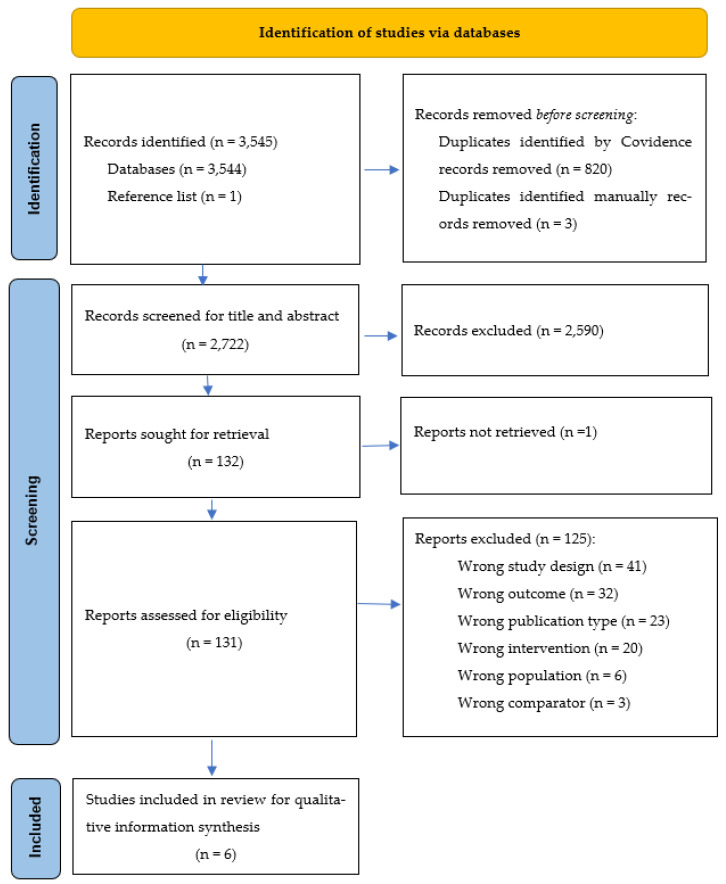
The search process according to PRISMA 2020 flow diagram [35].

**Table 1 healthcare-12-01542-t001:** Commonly used GV indices.

GV Index	Definition or Calculation Formula	Comments
Glucose SD	A measure of dispersion of measured glucose values around the mean.	Very familiar to physicians and easy to calculate [26].
Glucose CV	(SD)/(mean glucose)ǀ×ǀ100 [3]	A CV of ≤36% is recommended in patients with diabetes [27]. More robust than glucose SD because of taking into consideration the mean glucose values [3].
TIR	Percentage of time spent with the blood glucose levels within the range of 3.9–10 mmol/L	In general, it is recommended to be maintained above 70% [3,15]. Should be determined based on individual needs and risk of hypoglycaemia.
TAR	Percentage of time spent with blood glucose levels above 10 mmol/L [3]	Recommended targets differ depending on individual patients’ situations, but in general for patients with T2DM, a TAR > 10 mmol/L (level 1 hyperglycaemia) of <25%, and a TAR > 13.9 mmol/L (level 2 hyperglycaemia) of <5% is recommended [4].
TBR	Percentage of time spent with blood glucose levels below 3.9 mmol/L [3]	Recommended targets differ depending on individual patients’ situations, but in general for patients with T2DM, a TBR < 3.9 mmol/L (level 1 hypoglycaemia) of <4% and a TBR < 3.0 mmol/L (level 2 hypoglycaemia) of <1% is recommended [28].
MAGE	A measure of glucose fluctuations that exceed 1 SD from the mean (high and low) [3]	Takes into account peaks and nadirs of glucose levels not just the numbers of fluctuations [18]. Capable of identifying large glucose excursions.
MDD	Based on calculation of absolute differences between two glucose values measured 24 h apart [26]	A metric for estimating the between-day GV.
CONGA	SD of the differences of glucose readings for a defined period of time [26]	Measures within-day GV.
MAG	The sum of absolute differences among consecutively measured glucose values divided by the duration of time over which the measurements were conducted [29]	This is a good indicator of dramatic GV in critically ill patients, for example in intensive care units [30].
GMI	It is an estimated HbA1c based on the average glucose levels measured by CGM [3]	Needs glucose values for a consecutive period of 10–14 days for calculation [31].

CGM, continuous glucose monitoring; CV, coefficient of variation; CONGA, Continuous Overlapping Net Glycaemic Action; GMI, Glucose Management Indicator; GV, glycaemic variability; MAG, Mean Absolute Glucose; MAGE, Mean Amplitude of Glycaemic Excursions; MDD, Mean of Daily Differences; SD, standard deviation; TAR, Time Above Range; TBR, Time Below Range; TIR, Time in Range.

**Table 2 healthcare-12-01542-t002:** Characteristics of the included studies.

Authors (Year)	Study Design	Participants	Male, n (%)	Age, Mean (SD) or Median (IQR)	Duration of Diabetes, Years, Mean (SD) or Median (IQR)	GV Indices	HbA1c Comparator	Median Follow-Up Time	Outcome Measures
Lu et al. (2021) [40]	PCS	Adult patients with T2DM admitted to the hospital for diabetes management (n = 6225).	3404 (55%)	62 (12)	9.7 (7.4)	TIR and glucose CV	Yes	6.9 years	Cardiovascular mortality and all-cause mortality.
Su et al. (2018) [41]	PCS	Patients with T2DM and NSTE-ACS admitted to the hospital for elective PCI. Following measuring MAGE, they were divided into two groups (those with MAGE levels <3.9 mmol/L and those with MAGE ≥3.9 mmol/L) (n = 759).	177 (61%)	63 (10)	Mean duration between two groups: 25 months for patients with MAGE <3.9 mmol/L; 38 months for patients with MAGE ≥ 3.9 mmol/L.	MAGE	Yes	In-hospital period (not specified)	Primary outcome: in-hospital MACE including all-cause mortality, new-onset MI, AHF, and stroke. Secondary outcomes: Each of the components separately.
Mita et al. (2023) [42]	PCS	Outpatients with T2DM who did not have a symptomatic CVD at the inclusion (n = 600).	379 (63%)	65 (9)	11 (6–18)	TIR, TAR10, TAR13.9, TBR3.9, TBR3, glucose CV.	Yes	2 years	Glucose, CV, TIR, TAR and TBR.
Lu et al. (2021) [43]	PCS	Adult patients with T2DM admitted to the hospital for diabetes management (n = 6090).	3326 (55%)	62 (12)	10 (4–15)	Glucose CV, TIR, TAR7.8, TAR10, TAR13.9, TBR3.9, TBR3.	Yes	6.8 years	All-cause mortality.
Andersen et al. (2021) [44]	PCS	Insulin-treated patients with T2DM who had at least one microvascular complication (with or without macrovascular complications) (n = 21).	15 (71%)	67 (10)	18 (8)	Glucose CV; Glucose SD; TBR3.9; TBR3; TIR; TAR10; and TAR13.9.	Yes	12 months	Cardiac arrhythmias.
Ajjan et al. (2023) [45]	Open-label RCT	Patients with T2DM and acute MI who were receiving medications which potentially could cause hypoglycaemia (n = 141).	103 (73%)	63 (53–70)	13 (7.0–18.0)	TIR, TBR3.9, and TBR3.	Yes	3 months	Primary outcome measure: TIR on days 76–90. Secondary and exploratory outcome measures: hypoglycaemia, HbA1c, MACE, all-cause mortality, quality of life (QOL), and cost effectiveness.

ACS, acute coronary syndrome; AHF, acute heart failure; CVD, cardiovascular disease; GV, glycaemic variability; HbA1c, haemoglobin A1c; IQR, interquartile range; MACE, major adverse cardiac events; MAGE, mean amplitude of glycaemic excursions; MI, myocardial infarction; NSTE-ACS, non-ST segment elevation acute coronary syndrome; PCI, percutaneous coronary intervention; PCS, prospective cohort study; RCT, randomised controlled trial; SD, standard deviation; T2DM, type 2 diabetes mellitus; TAR, Time Above Range; TAR7.8, TAR > 7.8 mmol/L or >140 mg/dL; TAR10, TAR > 10 mmol/L or >180 mg/dL; TAR13.9, TAR > 13.9 mmol/L or >250 mg/dL; TBR, Time Below Range; TBR3.9 (TBR < 3.9 mmol/L or <70 mg/dL; TBR3, TBR < 3.0 mmol/L or <54 mg/dL; TIR, Time in Range.

**Table 3 healthcare-12-01542-t003:** The aims and outcomes of the included studies.

Authors (Year)	Title	Aim of the Study	Exposure or Intervention	Findings
Lu et al. (2021) [40]	Time in range in relation to all-cause and cardiovascular mortality in patients with type 2 diabetes: a prospective cohort study	To study the associations between TIR and all-cause mortality or CVD mortality in patients with T2DM.	Each patient wore a CGM sensor at the first day of hospitalisation, which was kept in place for three days.	There was a negative association between TIR and long-term risks of all-cause and CVD mortality. Increased risks of all-cause and CVD mortality were observed for patients with HbA1c < 6% and ≥8% compared with those with HbA1c levels between 6.0 and 6.9%. TIR was negatively associated with HbA1c levels, the history of CVD, and use of hypertension medicines, aspirin, and statins.
Su, et al. (2018) [41]	Admission glycaemic variability correlates with in-hospital outcomes in diabetic patients with non-ST segment elevation acute coronary syndrome undergoing percutaneous coronary intervention	To assess the relationship among admission GV with in-hospital MACE in patients with T2DM and NSTE-ACS undergoing PCI.	CGM devices were fitted after admission and monitored for 24–72 consecutive hours.	MACE occurred in 48 patients (6.3%) during hospital stay. The higher-MAGE group had higher rates of MACE (9.9% vs. 4.8%, *p* = 0.009) and all-cause mortality (2.3% vs. 0.4%, *p* = 0.023) compared with the lower-MAGE group. The rates of new-onset MI (1.9% vs. 1.4%), AHF (3.8% vs. 1.4%), and stroke (1.9% vs. 0.8%) were all higher in the high-MAGE than low-MAGE groups, but the changes were statistically non-significant (all *p* > 0.05). MAGE compared with HbA1c was a better predictor of in-hospital MACE (AUROC for MAGE = 0.608, 95% CI 0.524–0.692, *p* = 0.012; and for HbA1c = 0.556, 95% CI 0.475–0.637, *p* = 0.193).
Mita et al. (2023) [42]	Continuous glucose monitoring-derived time in range and CV are associated with altered tissue characteristics of the carotid artery wall in people with type 2 diabetes	Primary aim: to examine the relationship between TIR and CV at baseline, and changes in IMT and GSM. Secondary aim: to evaluate the association of other GV indices at baseline and HbA1c with changes in IMT and GSM.	Fitting of CGM (data extracted during the middle 8 days of the 14-day lifetime of CGM device) and ultrasound scan for carotid artery.	Over the study period of 104 weeks, IMT increased from 0.759 ± 0.153 mm to 0.773 ± 0.152 mm, *p* < 0.001. Similarly, thickened-lesion GSM increased from 43.5 ± 19.5 units to 53.9 ± 23.5 units, *p* < 0.001. However, no significant changes in common carotid artery maximum-IMT were observed (from 1.109 ± 0.442 to 1.116 ± 0.469 mm, *p* = 0.453). Baseline TIR and glucose CV were significantly associated with the annual change in thickened-lesion GSM. HbA1c was not associated with changes in the mean IMT, mean GSM or thickened-lesion GSM after adjusting for multiple testing.
Lu et al. (2021) [43]	Association of HbA1c with all-cause mortality across varying degrees of glycaemic variability in type 2 diabetes	To examine the relationships between HbA1c levels and all-cause mortality across different degrees of GV in patients with T2DM.	Each patient wore a CGM sensor at the first day of hospitalisation, which was kept in place for three days.	Amongst patients in the lowest and middle tertiles of glucose CV, an HbA1c ≥ 8.0% was associated with 136% (HR 2.36, 95% CI 1.46–3.81) and 92% (HR 1.92, 95% CI 1.22–3.03) increased risk of all-cause mortality, respectively. Each 10% decrease in TIR was associated with a 12% (HR 1.12, 95% CI 1.04–1.20) increase in the risk of all-cause mortality, and every 10% increase in TAR7.8, TAR10, and TAR13.9 was associated with an 11% (HR 1.11, 95% CI 1.04–1.19), 11% (HR 1.11, 95% CI 1.04–1.18), and 14% (HR 1.14, 95% CI 1.04–1.25) increase in the risk of all-cause mortality, respectively, among the patients in the highest tertile of glucose CV.
Andersen et al. (2021) [44]	Associations of hypoglycaemia, glycaemic variability, and risk of cardiac arrhythmias in insulin-treated patients with type 2 diabetes: a prospective, observational study	To examine the association between arrhythmia and glucose fluctuations and episodes of hypoglycaemia.	In months 1 and 12, patients wore a CGM sensor over four periods of six days; plus, one 6-day period of monitoring per month during the 10 months in between. CGM was undertaken for the total period of 108 days at maximum.	Time in hypoglycaemia was longer at night compared to daytime (median [IQR], 0.7% [0.7–2.7%] vs. 0.4% [0.2–0.8%] respectively). The severity of hypoglycaemia was slightly higher during daytime. Asymptomatic cardiac arrhythmias occurred more frequently at night compared to daytime (IRR 4.22 [3.48–5.15]). Incidence rate of arrhythmias had positive associations with CV and SD during night, but negative associations were observed during day. GV is an independent predictor of cardiac arrhythmias in patients with T2DM treated with insulin.
Ajjan et al. (2023) [45]	Multicentre randomised trial of intermittently scanned continuous glucose monitoring versus self-monitoring of blood glucose in individuals with type 2 diabetes and recent-onset acute myocardial infarction: results of the LIBERATES trial	To examine the impact of in-hospital use of isCGM in optimising glycaemic control and improving patient-related outcomes in persons with T2DM and recent MI.	In the intervention arm, participants wore isCGM sensors for 90 consecutive days and asked to change their sensors every 14 days. Patients in the control arm wore three sensors in total, two during the first month (14 days monitoring for each) and the third one on days 76–90.	In the intervention group, compared with the control (SMBG) group, TIR (days 76–90) increased by 17 min/day (95% credible interval −105 to +153 min/day) with 59% probability of benefit. When criteria were relaxed to include glucose coverage of 65% per day, the difference between study arms increased to 28 min in favour of the isCGM group. The isCGM group experienced lower hypoglycaemic exposure at days 76–90 (−80 min/day; 95% CI −118–−43) and days 16–30 (−28 min/day; 95% CI −92–2). Compared to baseline, both study arms showed similar reductions in HbA1c levels at 3 months (7 mmol/mol). Combined glycaemic emergencies and mortality occurred in four is-GM and seven SMBG participants.

AHF, acute heart failure; AUROC, area under the receiver operating characteristic curve; CGM, continuous glucose monitoring; CI, confidence interval; CVD, cardiovascular disease; GSM, grey-scale median; GV, glycaemic variability; HR, hazard ratio; IMT, intima–media thickness; IRR, incident rate ratio; IQR, interquartile range; isCGM, intermittently scanned continuous glucose monitoring; MACE, major adverse cardiac events; MI, myocardial infarction; NSTE-ACS, non-ST segment elevation acute coronary syndrome; PCI, percutaneous coronary intervention; SMBG, self-monitoring of blood glucose; T2DM, type 2 diabetes mellitus; TAR7.8, TAR > 7.8 mmol/L or >140 mg/dL; TAR10, TAR > 10 mmol/L or >180 mg/dL; TAR13.9, TAR > 13.9 mmol/L or >250 mg/dL; TIR, Time in Range.

**Table 4 healthcare-12-01542-t004:** Quality assessment for cohort studies based on the Newcastle–Ottawa Scale.

Study	Selection (Maximum 4 Stars)	Comparability (Maximum 2 Stars)	Outcome (Maximum 3 Stars)	Total Stars (out of Maximum 9)	Comments
Lu (2021) [40]	★★	★★	★★	6	Adjusted for several variables such as age, sex, BMI, diabetes duration, systolic blood pressure, smoking status, history of cancer and CVD, triglyceride, and HDL cholesterol. Blinded CGM measurement. No control group.
Su (2018) [41]	★★	★★	★	5	Adjusted for age ≥ 65 years, sex, BMI ≥ 30 kg/m^2^, HbA1c ≥ 7%, hypertension, and hyperlipidaemia.
Mita (2023) [42]	★★★	★★	★★	7	Adjusted for age, sex, and index values for carotid atherosclerosis. Of the 600 participants, 47 did not undergo carotid ultrasonography at 104 weeks, 35 were lost to follow-up and 12 refused to undergo the examination. No control group.
Lu (2021) [43]	★★	★★	★★	6	Adjusted for several variables such as diabetes duration, BMI, systolic blood pressure, triglyceride, and HDL cholesterol. Blinded CGM measurement. No control group.
Andersen (2021) [44]	★★	★	★★	5	Insufficient information about patient selection. The selected group of patients limits the applicability of the results. No control group. Very small cohort size.

## Data Availability

No new data were created or analyzed in this study.

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
