# Peer review of "Potential Benefits of Continuous Glucose Monitoring for Predicting Vascular Outcomes in Type 2 Diabetes: A Rapid Review of Primary Research"

_healthcare, 2024, doi:10.3390/healthcare12151542_

Round 1

Reviewer 1 Report (Previous Reviewer 1)

Comments and Suggestions for Authors

I believe the Manustript has been improved in terms of clarity. I still believe that both TIR and glucose variability are important metrics, but they provide different kinds of information.

Author Response

Comment: I believe the Manuscript has been improved in terms of clarity. I still believe that both TIR and glucose variability are important metrics, but they provide different kinds of information.

Response: We generally agree with your comment, and we appreciate it very much. However, to our understanding, glucose variability is an overarching concept which can be assessed via several metrics including but not limited to TIR, TAR, TBR as well as metrics which reflect the numbers and amplitudes of glucose peaks and nadirs over a defined period of time. Accordingly, TIR, not as a stand-alone index, but in conjunction with other glucose variability indices can provide us with information as to how well the glucose levels are maintained within the desirable range as opposed to deviating from it.

Reviewer 2 Report (Previous Reviewer 3)

Comments and Suggestions for Authors

I have no further comments.

Author Response

Comment: I have no further comments.

Response: Thank you very much for your comments. You helped us improve our manuscript.

Reviewer 3 Report (New Reviewer)

Comments and Suggestions for Authors

This manuscript appears to have undergone extensive revisions already, as the copy provided was heavily highlighted and select text was changed to red. The manuscript is well written and I have relatively minor revisions to request.

It is very noticeable that the authors are switching the spelling of glycemic (used 17x), glycaemic (used 22x), glycaemia (used 4x) and glycemia (used 1x). It needs to be decided on using the American or British spelling of these words and the manuscript needs to be edited so that these terms are consistent throughout the paper.

Abstract

Line 24: “More research is needed.” This statement needs to be expanded, and the details of larger study cohorts, longitudinal studies, or including diverse backgrounds could be added to this sentence.

Introduction

The introduction was incredibly well written and interesting.

Line 45: “or assessing glycaemic situations” needs to be changed to assess.

Line 50: HbA1c has the 1 in subscript, needs to be changed to regular text.

Methods

No comments

Results

No comments

Discussion

Line 375: “vascular complications have not been extensively studies” – needs to be changed to studied.

Conclusions

Well written.

Tables and Figures

No comments

References

No comments

Author Response

This manuscript appears to have undergone extensive revisions already, as the copy provided was heavily highlighted and select text was changed to red. The manuscript is well written and I have relatively minor revisions to request.

Comment 1. It is very noticeable that the authors are switching the spelling of glycemic (used 17x), glycaemic (used 22x), glycaemia (used 4x) and glycemia (used 1x). It needs to be decided on using the American or British spelling of these words and the manuscript needs to be edited so that these terms are consistent throughout the paper.

Response 1. Thank you very much for your attention. Although we had tried to use the British spelling consistently, but we had used American spelling mistakenly on a few occasions which you rightfully helped us notice and fix them all (i. e., to convert them to British spelling). However, please notice that many of those cases you referred to, are in fact in the reference list and they are the spelling used by the published articles. Therefore, we believe that we should not change the spelling of the reference entries to maintain the originality of the references’ details.

Comment 2. Abstract - Line 24: “More research is needed.” This statement needs to be expanded, and the details of larger study cohorts, longitudinal studies, or including diverse backgrounds could be added to this sentence.

Response 2. Thanks for the comment. To address this, we replaced the above-mentioned sentence in the Abstract with “There is a need for longitudinal studies in larger and more diverse populations, longer follow-ups, and use of numerous CGM-derived GV indices while collecting information about all microvascular and macrovascular complications”.

Comment 3. Introduction - The introduction was incredibly well written and interesting.

Line 45: “or assessing glycaemic situations” needs to be changed to assess.

Line 50: HbA1c has the 1 in subscript, needs to be changed to regular text.

Response 3. Both changes were implemented (and highlighted in yellow). Thank you. 

Comment 4. Methods - No comments

Comment 5. Results - No comments

Comment 6. Discussion - Line 375: “vascular complications have not been extensively studies” – needs to be changed to studied.

Response 6. We amended this. Thank you.

Comment 7. Conclusions - Well written.

Comment 8. Tables and Figures - No comments

Comment 9. References - No comments

Reviewer 4 Report (New Reviewer)

Comments and Suggestions for Authors

Limited but significant evidence suggests that GV indices may predict clinical complications in T2DM, both in the short and long term. 

The main limitation of this study is its very few number of studies included.

Specify how CGM differs in its adoption between type 1 and type 2 diabetes management.

"Rapid literature review" : why the authors choosed a rapid literature review ?

Clarify the criteria for excluding lower-quality studies.

Clarify what is meant by "vascular complications"

Editing spell and grammar is necessary

Comments on the Quality of English Language

minor editing

Author Response

Limited but significant evidence suggests that GV indices may predict clinical complications in T2DM, both in the short and long term. 

Comment 1. The main limitation of this study is its very few number of studies included.

Response 1. We agree with this comment, and we have mentioned this in our study and that our findings should be interpreted cautiously because of the limited number of studies been undertaken in relatively small cohorts.

Comment 2. Specify how CGM differs in its adoption between type 1 and type 2 diabetes management.

Response 2. The evidence suggests that CGM technology is more widely adapted in people with type 1 diabetes mellitus (T1DM) compared with people with type 2 diabetes mellitus (T2DM). Some of the reasons include, but they may not be limited to: people with T1DM are younger and more tech savvy, they usually have more intensive insulin regimens (which increases the risk of hypoglycaemia) and have higher HbA1c levels. In addition, insurance companies are more willing to subsidise CGM devices in T1DM compared with T2DM (because of less evidence in favour of the effectiveness of this technology in T2DM). The other factor is that increased adaptation of the CGM in T1DM is beneficial in combining it with insulin pumps to make closed loop systems leading to more efficient glucose control (e. g., in building a bionic pancreas). These very interesting topics, but we think they are out of the scope of our research (at this stage) to go into the depth of the matter. The references we have used to answer this comment are as follows: https://www.ncbi.nlm.nih.gov/pmc/articles/PMC10228889, https://www.ncbi.nlm.nih.gov/pmc/articles/PMC6440103, and https://www.ncbi.nlm.nih.gov/pmc/articles/PMC9207329

Comment 3. "Rapid literature review" : why the authors choose a rapid literature review ?

Response 3. Rapid reviews are becoming increasingly popular in medical research due to numerous factors such as their efficiency of providing evidence with fewer numbers of researchers in shorter amounts of time compared with systematic reviews while sharing many critical components with systematic reviews such as, but not limited to using a specifically defined narrow research question, involvement of more than one reviewers and systematic analyses of the quality of evidence found (https://ebm.bmj.com/content/early/2024/03/14/bmjebm-2023-112722). In our case, before deciding about doing this rapid review, we had noticed the scarcity of evidence about the effectiveness of CGM for treatment of type 2 diabetes mellitus. This is reflected by the lack or the existence of weak recommendations about the use of such methods in clinical guidelines for T2DM management. Hence, given the huge health burdens imposed by T2DM (and the urgency of the matter), we decided that instead of doing a systematic review which could take 1 – 2 years to complete to find a clue to this matter, we can do a rapid review in around 6 months to find evidence to provide to clinical decision makers, or at least to find some literature gaps which we could focus on in mapping out our future research in this area. Accordingly, we think that this rapid review has been successful in providing evidence in both areas, especially in identifying areas in need for future research.

Comment 4. Clarify the criteria for excluding lower-quality studies.

Response 4. Thanks for your comment. Firstly, we all know that the range of quality of evidence in published studies is a spectrum and not a dichotomised classification (i. e., high quality versus low quality evidence) and that assessments by different researchers could have slight subjectivity (knowing that the guidelines are for guidance not for strict ruling). Secondly, we addressed your comment by amending a sentence Under the 2.2.3. Type of Studies, which now reads as follows: “Other study designs such as cross-sectional studies, case series, case reports, case studies, and expert opinion papers were all excluded due to being generally considered of lower quality”. In our decision to do so, we used the guidelines for levels of evidence issued by the Australian National Institute of Clinical Studies (chrome-extension://efaidnbmnnnibpcajpcglclefindmkaj/https://www.nhmrc.gov.au/sites/default/files/images/appendix-f-levels-of-evidence.pdf). Accordingly, we inserted a new reference for this in the text (reference number 37).

Comment 5. Clarify what is meant by "vascular complications"

Response 5. As mentioned in the Introduction section, by the term “vascular complications”, we mean a wide range of morbidities including macrovascular (ischaemic heart disease, stroke, and peripheral artery disease) and microvascular (nephropathy, neuropathy, and retinopathy) diseases and cardiac arrhythmias.

Comment 6. Editing spell and grammar is necessary

Response 6. Thanks for your comments. We checked our manuscript for spelling and grammar one more time and we improved it accordingly.

This manuscript is a resubmission of an earlier submission. The following is a list of the peer review reports and author responses from that submission.

Round 1

Reviewer 1 Report

Comments and Suggestions for Authors

This study aimed to find evidence about the use of CGM in patients with T2DM and its value in predicting vascular complications. The review clearly states the limited research on the direct relationship between glucose fluctuations and vascular complications in T2DM patients, however, some confusing sections need major revision. Accordingly, the Title needs adjustment.

The main concern is that while Time in Range (TIR) offers valuable insight into glucose variability, it presents a distinct perspective compared to traditional measures of GV like standard deviation (SD) and coefficient of variation (CV). TIR quantifies the percentage of time a patient's blood glucose remains within a pre-defined target range, Higher TIR values DO NOT directly indicate greater glucose variability. SD and CV are statistical metrics that assess the dispersion of blood glucose readings around the mean. Higher values indicate greater glucose variability.

I would suggest excluding the sections commenting on TIR, TBR, and TAR in the context of assessing glucose variability.

The aim is badly defined in the Abstract, it should follow the research objectives …..To explore associations between blood glucose excursions (measured by CGM) and short- or long-term vascular complications in patients with T2DM.

Furthermore, in the Abstract: the authors should note that only high-level evidence excluding cross-sectional was reviewed.

Strengths:

Transparent reporting: The authors clearly outline the protocol, search strategy, eligibility criteria, and data extraction process following PRISMA guidelines.

Comprehensive search: The inclusion of Cochrane Library, MEDLINE, and Scopus ensures a thorough search of relevant literature.

Eligibility criteria: The criteria are clear and appropriate, focusing on studies with T2DM participants, CGM intervention, and clinically relevant outcomes.

Risk of bias assessment: The use of established tools (NOS for observational studies and Cochrane RoB 2 for RCTs) demonstrates a commitment to evaluating the quality of included studies.

Data extraction: Double-reviewer extraction with a third reviewer for cross-checking minimizes errors and ensures data accuracy.

Specific examples: It provides specific details on the GV indices used in the studies and highlights some interesting associations observed with CV, SD, and MAGE.

Limitations discussed: The limitations of extrapolating findings from a single high-quality RCT to wider T2DM populations are acknowledged.

Suggestions:

Search strategy details: While a full record is included in a supplementary table, mentioning the specific search terms used (even a few examples) could provide further transparency.

CGM discussion: The section addresses the lack of studies investigating CGM's impact on long-term vascular outcomes and compares it with HbA1c.

In my opinion, the following objectives were too broad - to analyze the existing knowledge and identify knowledge gaps about the usefulness of CGM in patients with T2DM in improving vascular outcomes. Objective 2: To compare the short or long-term prognosis for patients with T2DM whose treatments were informed by CGM data, with those patients whose diabetes management protocols were devised based on the use of HbA1c only.

Overall, parts of the manuscript are well-written and present significant findings. The goal is to make this research as accessible and understandable as possible to a broad audience – following a revision.

Reviewer 2 Report

Comments and Suggestions for Authors

1. In lines 139-140, the author states, "Studies including persons 18 years of age or older with T2DM who were or were not administered insulin were included. Studies including only persons with T1DM mellitus or gestational diabetes, studies undertaken in children, and studies done on animals." Therefore, did the author also collect data on T2DM, T1DM, and GDM?

2. In lines 140-155, why did the author not include studies related to glycated proteins other than HbA1c?

3. For this systematic review, did the author register with PROSPERO or any other relevant registry before beginning the research?

4. The author included a wide variety of articles, even case reports and case series, but did not specify the exclusion criteria. Moreover, it is unclear which type of study the final six articles selected belong to.

5. The author seems to intend to demonstrate the predictive power of GV (Glycemic Variability), but it appears that without conducting a meta

Reviewer 3 Report

Comments and Suggestions for Authors

I enjoy reading this review which focuses on the association between GV and complication in T2D. There are a few minor corrections:

1. page 16, line 307-308, the unit and value of time in range look wrong. the unit of TIR should be percentage, not mmol/L

2. in Table 1, the description of MAGE can be a bit more accurate and specific. the recommended target for TAR, TBR should be commented.

3. page 18, line 406-407, reference 44 is about hypoglycemia, not glucose fluctuations (CV, SD). Do we have evidence/study showing excessive glucose fluctuation "cause" vascular disease? 

Reviewer 4 Report

Comments and Suggestions for Authors

The authors made an intriguing attempt to show the benefit of CGM use in people with type 2 diabetes by reviewing the six research reports extracted from the database sets. There are several points to be addressed by the authors.

First, GMI is supposed to stay among indices of glycemic variability in Table 1?

Second, since the authors tried to compare the outcomes between with and without CGM, the second objective of the study mentioned before the section of Materials and Methods seemed to be inappropriate: “To compare --------, with those patients whose diabetes management protocols were devised based on the use of HbA1c only”.

Third, if the study that used HbA1c as a surrogate measure for clinical outcomes, but did not directly measure clinical vascular outcomes was excluded, why the report of Ajjan et al. was included in the present review? The authors admitted in Discussion like “given the specifications of the included participants it is unclear that the results of the study can be extrapolated to the wider groups of patients with T2DM”. By the way, the detailed data from the study of Ajjan et al. was not displayed in Supplementary Table 2.

Fourth, the authors should describe the logical reasons why 2 articles by Lu et al. were both included and HbA1c was presented in the report (2021b) but not in (2021a) even though they were articles of a single large study (the INDices of continuous Glucose monitoring and adverse Outcomes of diabetes (INDIGO) study).

Fifth, the first paragraph of page 15: “The quality of the only randomised clinical trial was determined as high, using RoB 2 (see Supplementary Table 3). However, this study did not have the sufficient power to effectively assess the vascular outcomes.” is in the right place of the manuscript? That seemed to be out of context.
